# Molecular mechanisms of gating in the calcium-activated chloride channel bestrophin

Alexandria N Miller[1†], George Vaisey[1,2†], Stephen B Long[1]*

[1]Structural Biology Program, Memorial Sloan Kettering Cancer Center, New York, United States; [2]Louis V. Gerstner Jr. Graduate School of Biomedical Sciences, Memorial Sloan Kettering Cancer Center, New York, United States

**Abstract** Bestrophin (BEST1-4) ligand-gated chloride ($Cl^-$) channels are activated by calcium ($Ca^{2+}$). Mutation of BEST1 causes retinal disease. Partly because bestrophin channels have no sequence or structural similarity to other ion channels, the molecular mechanisms underlying gating are unknown. Here, we present a series of cryo-electron microscopy structures of chicken BEST1, determined at 3.1 Å resolution or better, that represent the channel's principal gating states. Unlike other channels, opening of the pore is due to the repositioning of tethered pore-lining helices within a surrounding protein shell that dramatically widens a neck of the pore through a concertina of amino acid rearrangements. The neck serves as both the activation and the inactivation gate. $Ca^{2+}$ binding instigates opening of the neck through allosteric means whereas inactivation peptide binding induces closing. An aperture within the otherwise wide pore controls anion permeability. The studies define a new molecular paradigm for gating among ligand-gated ion channels.
DOI: https://doi.org/10.7554/eLife.43231.001

*For correspondence:
longs@mskcc.org

†These authors contributed equally to this work

**Competing interests:** The authors declare that no competing interests exist.

## Introduction

The family of bestrophin proteins (BEST1-4) was identified by linkage analysis to hereditary macular degenerations caused by mutations in BEST1 (*Petrukhin et al., 1998*; *Marquardt et al., 1998*). To date more than 200 mutations in BEST1 are linked with eye disease (*Johnson et al., 2017*; *Xiao et al., 2010*). BEST1-4 proteins are expressed in the plasma membrane and form $Ca^{2+}$-activated $Cl^-$ channels by assembling as pentamers (*Hartzell et al., 2008*; *Kane Dickson et al., 2014*; *Vaisey et al., 2016*; *Sun et al., 2002*; *Tsunenari et al., 2006*). Data indicate that BEST1 mediates a $Ca^{2+}$-activated $Cl^-$ current that is integral to human retinal pigment epithelial function (*Li et al., 2017*). The broad tissue distribution of the family suggests additional physiological functions that are not fully realized (*Hartzell et al., 2008*), and these may include processes as diverse as cell volume regulation (*Fischmeister and Hartzell, 2005*; *Milenkovic et al., 2015*), pH homeostasis (*Yu et al., 2010*), and neurotransmitter release (*Lee et al., 2010*).

An X-ray structure of chicken BEST1, which shares 74% sequence identity with human BEST1 and possesses analogous $Ca^{2+}$-activation and anion-selectivity properties, revealed an architecture that is distinct from other ion channel families (*Kane Dickson et al., 2014*; *Vaisey et al., 2016*). The channel comprises five BEST1 subunits arranged symmetrically around a central ion pore that is ~95 Å long. The diameter of the pore varies along its length and contains two constrictions, the 'neck' and the 'aperture', both of which are lined by hydrophobic amino acids. The X-ray structure of a bestrophin channel from the prokaryotic organism *Klebsiella pneumoniae* (KpBest) revealed a similar architecture, including analogous neck and aperture regions, in spite of minimal (14%) sequence identity to human BEST1 (*Yang et al., 2014*). Surprisingly, however, whereas metazoan bestrophin channels

are anion-selective and activated by $Ca^{2+}$, KpBest is cation-selective and does not contain the $Ca^{2+}$ sensor region (the 'Ca$^{2+}$ clasp') that is conserved amongst metazoan organisms.

From electrophysiological studies of purified chicken BEST1 and of mutants inspired by the structure, it was deduced that $Ca^{2+}$-activation of the channel is due to the binding of $Ca^{2+}$ ions to the cytosolic $Ca^{2+}$ clasps, of which there are five owing to the pentameric architecture, and to subsequent conformational changes in the neck (*Vaisey et al., 2016*). These experiments made it clear that the neck forms the activation gate that opens in response to the binding of cytosolic $Ca^{2+}$ ions. Perplexingly, however, even though the $Ca^{2+}$ clasps were occupied by $Ca^{2+}$ ions in the X-ray structure, the conformation of the neck was too narrow to allow hydrated ions to pass. We initially postulated that phenylalanine residues within the neck (F80 and F84) might coordinate permeating anions via anion-quadropole interactions (also referred to as anion-π interactions) and thereby stabilize dehydrated anions as they passed though a narrow neck (*Kane Dickson et al., 2014*). However, we found that mutation of these residues to alanine had no effect on the ion selectivity properties of the channel, which made it seem unlikely that permeation through the neck was facilitated by anion-quadropole interactions in the wild type channel (*Vaisey et al., 2016*). Further, molecular simulation studies suggested that the conformation of the neck observed in the X-ray structure would form a significant energetic barrier that would likely prevent ion permeation (*Rao et al., 2017*). Thus, it was not clear what conformation the X-ray structure represented or how anions might be able to pass through the neck.

Our recent electrophysiological studies using purified chicken BEST1 in synthetic bilayers show that, like many ion channels, BEST1 undergoes inactivation wherein ionic currents through BEST1 decrease over time (*Vaisey and Long, 2018*). We now realize that the 'current rundown' previously observed for mammalian bestrophin channels is due to channel inactivation and that channel inactivation is an inherent property of bestrophin channels and not due to other cellular factors. Inactivation is stimulated by high (>500 nM) concentrations of $Ca^{2+}$ and is caused by the binding of a C-terminal 'inactivation peptide', contained within amino acids 346–367, to a receptor on the channel's cytosolic surface that is located near the $Ca^{2+}$ clasp (*Vaisey and Long, 2018*). Inactivation can be substantially diminished by mutation of the receptor site or mutation of the inactivation peptide, and it can be prevented altogether by removal of the inactivation peptide (*Vaisey and Long, 2018*). From these and other mutagenesis studies it was additionally concluded that the neck functions as the inactivation gate that prevents ion flow in the inactivated state (*Vaisey and Long, 2018*). These results began to suggest that the X-ray structure might represent an inactivated conformation of the channel.

The X-ray structure of BEST1 was determined in complex with an antibody (denoted 10D10) to facilitate crystallization. Electrophysiological analyses of the antibody's effect on BEST1 currents recorded using purified channels that were reconstituted into lipid bilayers indicate that it stimulates inactivation (*Vaisey and Long, 2018*), providing further evidence that the X-ray structure represents an inactivated state of the channel. Thus, because the inactivation peptide is bound to its receptor in the X-ray structure and because the antibody used for co-crystallization stimulates inactivation, we suspect that the X-ray structure represents an inactivated state in which the neck is too narrow to allow ions to flow through it (*Kane Dickson et al., 2014*; *Vaisey and Long, 2018*). The aforementioned studies suggest that the neck must widen to allow ion flow, but the extent of this widening and the conformational changes that underlie it are not clear. For example, minor movements of the side chain residues lining the neck might be sufficient for ion conduction, or opening might involve more dramatic rearrangements. Another unanswered question is how $Ca^{2+}$ binding controls the neck. It is also unclear what conformation the channel might adopt when the $Ca^{2+}$ clasps are emptied of their $Ca^{2+}$ ions. To address these questions we determined a series of cryo-electron microscopy (cryo-EM) structures of chicken BEST1 with and without $Ca^{2+}$ in activated, deactivated, and inactivated states that represent the major gating transitions of the channel. We find that ion conduction is permitted by dramatic widening of the neck and by correspondingly dramatic conformational changes in the protein. The conformational changes are more akin to localized protein refolding rather than the twisting or domain movement changes that typically underlie gating in ion channels and define the molecular bases of activation and inactivation gating in bestrophin channels.

## Results

To begin studying the structural basis of gating in BEST1, we determined a 3.1 Å resolution cryo-EM structure of $Ca^{2+}$-bound BEST1 without an antibody (the same construct used for X-ray studies, comprising residues 1–405 of chicken BEST1, and termed $BEST1_{405}$). From single particle analysis we obtained only a single conformation, which indicates substantial conformational homogeneity among the ~300,000 channel molecules in the dataset. The conformation of the channel is indistinguishable from the previously published X-ray structure (*Kane Dickson et al., 2014*) (*Figure 1—figure supplements 1,2*, RMSD for Cα atoms = 0.5 Å). As in the X-ray structure, each of the inactivation peptides, of which there are five owing to the symmetry of the channel, is bound to its cytosolic receptor and well defined in the density. Therefore, this cryo-EM structure also represents a $Ca^{2+}$-bound inactivated state. This result indicates that antibody binding does not distort the channel from a native conformation and reflects a satisfying correspondence of structures determined using X-ray crystallography and cryo-EM methodologies.

### An open channel conformation

In an aim to obtain structural information on an open conformation of the channel, we used a construct spanning amino acids 1–345 (termed $BEST1_{345}$) for cryo-EM studies. $BEST1_{345}$ does not inactivate because it is missing the C-terminal inactivation peptide (*Vaisey and Long, 2018*). Importantly, whilst it does not inactivate, we have previously shown that $BEST1_{345}$ possesses normal Cl$^-$ selectivity and $Ca^{2+}$-dependent activation (*Vaisey and Long, 2018*). Single particle cryo-EM analysis of $Ca^{2+}$-bound $BEST1_{345}$ (*Figure 1—figure supplements 2,3*) revealed two distinct conformations of the channel (*Figure 1*, *Figure 1—figure supplement 4*). The first, determined at 3.0 Å resolution, represents 86% of the particles and is essentially indistinguishable from the structure of $BEST1_{405}$, except for the absence of the inactivation peptide (*Figure 1A,C*, *Figure 1—figure supplement 1C*, RMSD for Cα atoms = 0.5 Å). Because $BEST1_{345}$ does not inactivate, the structure presumably represents a $Ca^{2+}$-bound closed conformation. The conformation, in which the activation gate (the neck) is closed even though $Ca^{2+}$ is bound, is consistent with single channel recordings of *drosophila* BEST1, which indicate that the channel has a low probability of being open (*Chien et al., 2006*). The neck adopts the same conformation in both the $Ca^{2+}$-bound closed structure and the inactivated structure, (*Figure 2C*). The second conformation present within the cryo-EM dataset, which represents 14% of the particles and is determined to 2.9 Å resolution, contains a dramatically widened pore within the neck (*Figure 1B,D*). Based on discussions presented herein we conclude that this represents the open conformation of the channel. The relative abundance of the closed conformation suggests that it is energetically favorable.

In the closed conformation, the neck is less than 3.5 Å in diameter and approximately 15 Å long; three hydrophobic amino acids on the neck helix (S2b) from each of the channel's five subunits, I76, F80 and F84, form its walls (*Figure 1C,E,F*) (*Kane Dickson et al., 2014*). Its narrowness, length and hydrophobicity would create an energetic barrier to ion permeation, consistent with simulation studies of wild-type and neck mutant BEST1 channel structures (*Rao et al., 2017*). In the open conformation, the neck has dilated to approximately 13 Å in diameter, which is more than sufficient to allow permeation of hydrated Cl$^-$ ions (*Figure 1E*). No appreciable conformational difference is present in the cytosolic region of the channel, and in particular, the aperture constriction of the pore retains its dimensions. In accord with conclusions drawn from previous electrophysiological studies of mutations made within the neck and aperture (*Vaisey et al., 2016*; *Vaisey and Long, 2018*), we conclude that the neck functions as a gate. It permits ion flow in the open conformation and prevents it in the closed conformation.

### Opening transition of the neck

*Figure 2* and *Figure 2—video 1* depict the transition between the closed and open conformations of the neck. Opening is reminiscent of the dilation of the iris of a camera lens. Conformational changes in the helices that form the walls of the neck (the S2b 'neck' helices from each of the five subunits) underlie opening. An unusual structural element allows for the movement of the neck helices and distinguishes the mechanism of gating in BEST1 from most other channels. Unlike most ion channels in which transmembrane segments are entirely α-helical, each S2b 'neck' helix is flanked on both of its ends by loop-like regions of extended secondary structure that partition what would

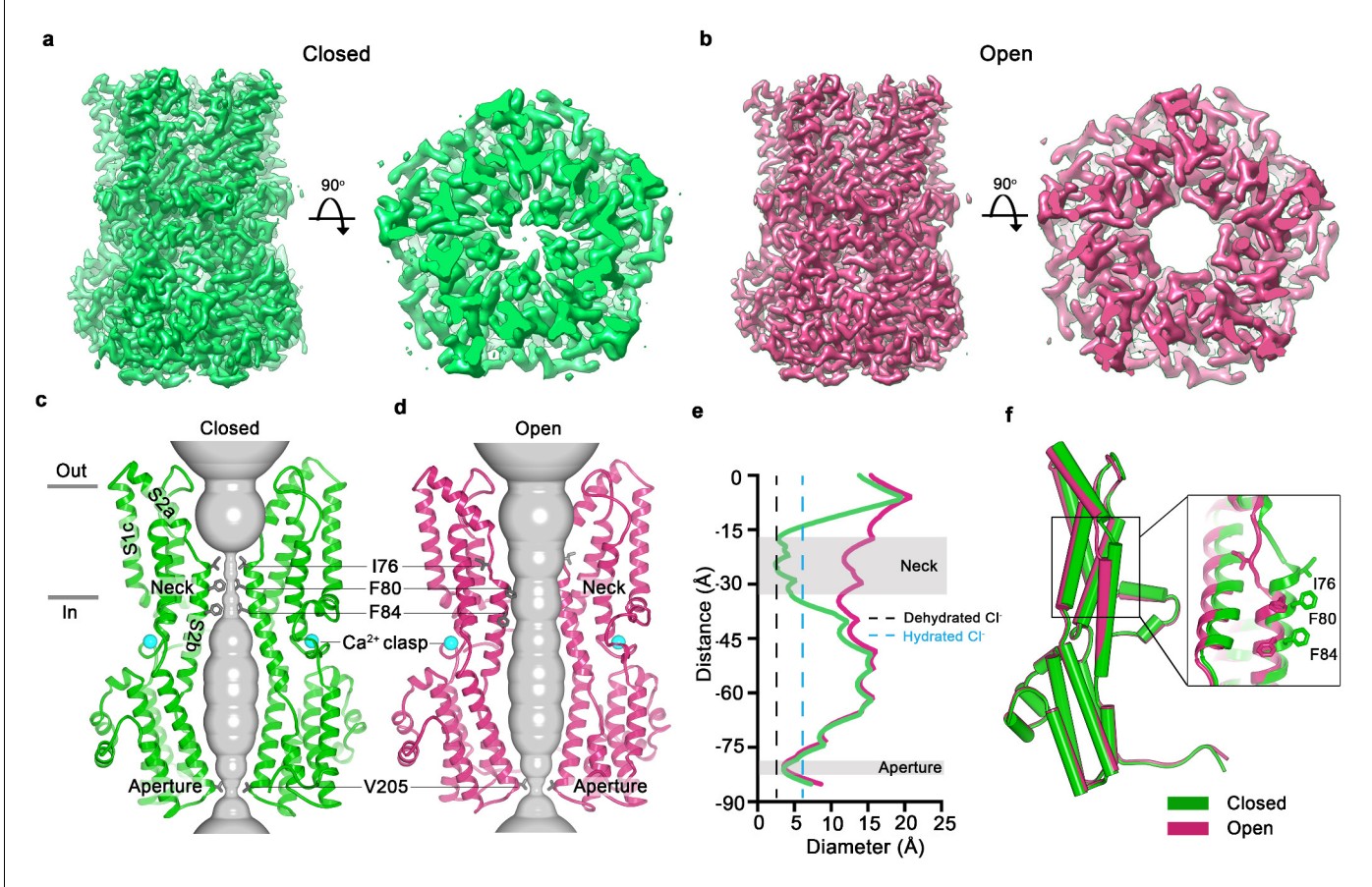

**Figure 1.** Open and closed conformations. (**a–b**) Cryo-EM maps of the $Ca^{2+}$-bound closed (green; non-conductive) and $Ca^{2+}$-bound open (pink) conformations of $BEST1_{345}$, depicted from the side (left panels) and as slices of the neck region viewed from the extracellular side (right panels). (**c–d**) Cutaway views of the $Ca^{2+}$-bound closed and $Ca^{2+}$-bound open conformations of $BEST1_{345}$. The pore (grey surface) is depicted as minimal radial distance from its center to the nearest van der Waals protein contact. Two subunits are drawn as ribbons; three are omitted for clarity. Amino acids in the neck and aperture regions are drawn as gray sticks; $Ca^{2+}$ ions are cyan spheres. Approximate boundaries of the lipid membrane are indicated by horizontal bars. (**e**) Pore dimensions in the open and closed conformations. Dashed lines indicate the diameters of a dehydrated (black) and hydrated (cyan) $Cl^-$ ion. (**f**) Superposition of individual subunits from the closed and open conformations with α-helices depicted as cylinders. The boxed area shows a close-up of the neck region, with α-helices depicted as cartoons and side chains as sticks.

DOI: https://doi.org/10.7554/eLife.43231.002

The following figure supplements are available for figure 1:

**Figure supplement 1.** Cryo-EM workflow for the $BEST1_{405}$ $Ca^{2+}$-bound dataset and comparison of EM and X-ray structures.
DOI: https://doi.org/10.7554/eLife.43231.003
**Figure supplement 2.** Structure determination of: $Ca^{2+}$-bound $BEST1_{405}$ (inactivated), $Ca^{2+}$-bound open $BEST1_{345}$, and $Ca^{2+}$-bound closed $BEST1_{345}$.
DOI: https://doi.org/10.7554/eLife.43231.005
**Figure supplement 3.** Cryo-EM workflow for the $BEST1_{345}$ $Ca^{2+}$-bound dataset.
DOI: https://doi.org/10.7554/eLife.43231.004
**Figure supplement 4.** Cutaway views of the $Ca^{2+}$-bound closed (left) and $Ca^{2+}$-bound open (right) conformations of $BEST1_{345}$.
DOI: https://doi.org/10.7554/eLife.43231.006
**Figure supplement 5.** Representative cryo-EM density for three BEST1 cryo-EM structures.
DOI: https://doi.org/10.7554/eLife.43231.007

otherwise be a continuous α-helix into three segments – S2a, S2b and S2c (*Figure 2C,D*). From comparison of the open and closed conformations of the neck, it is apparent that these loop-like regions provide a degree of flexibility that allows the neck helices to adopt their open and closed conformations and also provide a degree of tethering to the ends of these helices. Consequently, while the

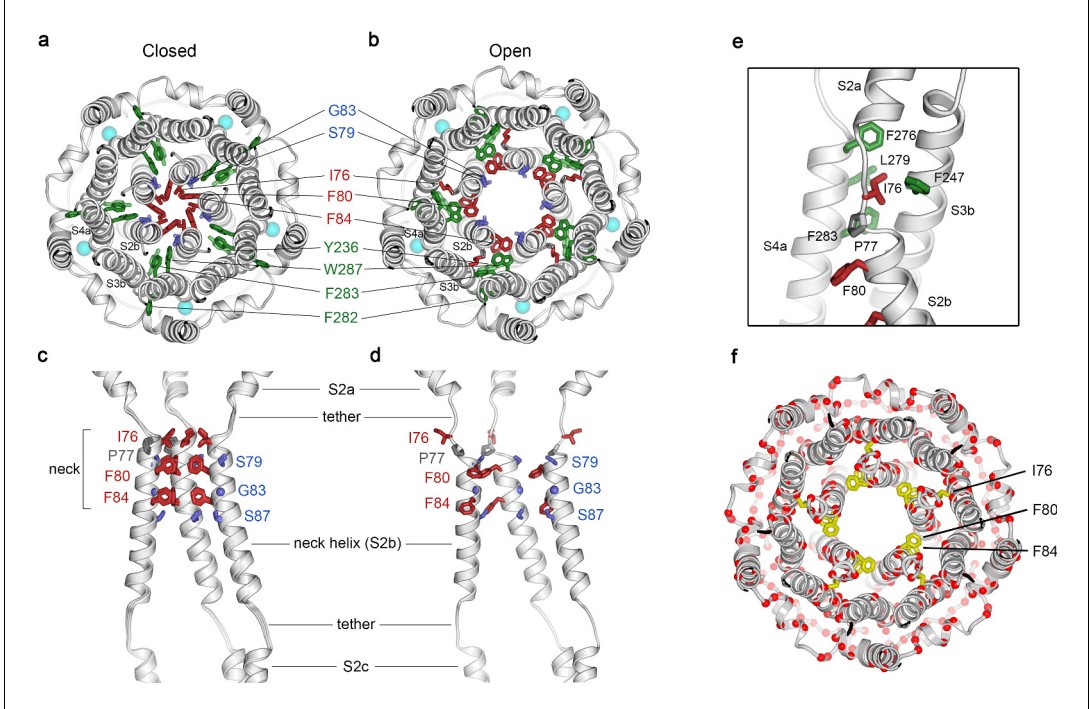

**Figure 2.** Opening transitions. (**a–b**) Cutaway views of the neck region for the closed (**a**) and open (**b**) conformations, viewed from the extracellular side and shown as ribbons. Residues that form the hydrophobic seal in the closed conformation (I76, F80, F84) are colored red in both conformations. Surrounding aromatic residues that move to accommodate opening are colored green. Residues that become exposed to the pore in the open conformation (S79, sticks, and G83, sphere) are blue. Cyan spheres represent $Ca^{2+}$ ions. A supplementary video shows the transition. (**c–d**) Side view of the conformational changes in the neck; closed (**c**) and open (**d**). In (**c**), a superposition of the structures of BEST1$_{405}$ in the $Ca^{2+}$-bound inactivated conformation, BEST1$_{345}$ in the $Ca^{2+}$-bound closed conformation, and BEST1$_{345}$ in the $Ca^{2+}$-free closed conformation shows that the neck adopts an indistinguishable (closed) conformation in each. The S2a,b, and c helices from three subunits are shown. Residues are depicted and colored as in a-b; P77 is gray; S87 is shown for reference. (**e**) A close-up view showing the hydrophobic packing of I76 in the open conformation. Neck residues are highlighted in red, neighboring hydrophobic residues that interact with I76 are shown in green, and P77 is depicted in gray. (**f**) Location of missense mutations associated with retinal diseases (*Johnson et al., 2017*; *Xiao et al., 2010*) at amino acid positions near the neck (red spheres indicate the Cα positions of the mutations).

DOI: https://doi.org/10.7554/eLife.43231.008

The following video and figure supplement are available for figure 2:

**Figure supplement 1.** Sidechain movements at the neck.

DOI: https://doi.org/10.7554/eLife.43231.009

**Figure 2–video 1.** Video showing opening transitions of the neck.

DOI: https://doi.org/10.7554/eLife.43231.010

neck helices undergo substantial movement, the S2a and S2c helices remain essentially fixed during the gating transition.

In the closed conformation of the neck, hydrophobic packing at the center of the neck among the I76, F80 and F84 residues themselves stabilize this conformation. In the open conformation, the tendency of the phenylalanine residues to seclude their hydrophobicity from an aqueous environment is satisfied by their interactions with other hydrophobic amino acids (Y236 and W287) on the S3b and S4a helices located behind the neck helix (*Figure 2B*). The conformational change moves F80 and F84 away from the center of the pore and involves rotation of the neck helix along its helical axis (~15 ° clockwise viewed from the extracellular side) and outward displacement of the neck helix that is more pronounced on its N-terminal end (~2.5 Å at F80). The opening transition includes a slight expansion of the entire transmembrane region of the channel (~1 Å increase in radius), and a coordinated set of side chain rotamer changes (*Figure 2A,B*, *Figure 2—video 1*). Both F80 and F84 move from the most commonly observed rotamer for phenylalanine (observed 44% of the time in the PDB) in the closed conformation to the second most commonly-observed rotamer (observed 33% of the

time) in the open conformation (*Lovell et al., 2000*). By these conformational changes, F80 and F84 rotate away from the axis of the pore (by 80° and 105°, respectively, *Figure 2*, *Figure 2—figure supplement 1*). Accompanying side chain rotamer changes that occur among neighboring residues (Y236, F282, F283, and W287) allow for the movements of F80 and F84 (*Figure 2A,B*, *Figure 2— video 1*). The conformational change in I76 is also dramatic. When the channel opens, the first α-helical turn of the neck helix unravels such that I76 packs with F247, F276, L279 and F283 in the open conformation and has shifted by approximately 10 Å (*Figure 2E*, *Figure 2—figure supplement 1*). The unraveling is facilitated by P77, which is perfectly conserved among BEST channels and is part of the neck helix in its closed conformation, but marks its N-terminal end in the open conformation (*Figure 2C,D*). The repositioning within the neck also exposes the hydrophilic residues S79 and G83 on the neck helix, which had been secluded behind the F80 and F84, to the pore in its open conformation (*Figure 2A,B*, *Figure 2—video 1*). Thus, through a concertina of coordinated conformational changes in and around the neck helix, amino acids that formed the hydrophobic barrier that prevented ion permeation have dispersed and reveal a wide aqueous vestibule, the diameter of which would be large enough to allow permeation of hydrated halide ions and larger molecules as well.

## Ca$^{2+}$-free conformation

To address how Ca$^{2+}$ binding contributes to BEST1 channel gating, we determined the cryo-EM structure of BEST1$_{345}$ in the absence of Ca$^{2+}$ (*Figure 3*, *Figure 3—figure supplements 1* and *2*). The structure is determined to 3.0 Å resolution and is indistinguishable from a cryo-EM structure of Ca$^{2+}$-free BEST1$_{405}$ at 3.6 Å resolution. As was expected from biochemical and electrophysiological analyses (*Vaisey and Long, 2018*), the inactivation peptide is not bound to its receptor in the structure of BEST1$_{405}$ without Ca$^{2+}$ (*Figure 3—figure supplement 3*). The neck is closed in the absence of Ca$^{2+}$ (*Figure 3A*), and notably, it shares the same closed conformation as in the inactivated and Ca$^{2+}$-bound closed structures (*Figure 2C*). The overall structure of the channel is very similar to the inactivated and Ca$^{2+}$-bound closed conformations with the only notable differences near the Ca$^{2+}$ clasp. In structures with Ca$^{2+}$ bound, the five Ca$^{2+}$ clasps, one from each subunit, resemble a belt that wraps around the midsection of BEST1 (*Figure 1A,B*). Without Ca$^{2+}$, the majority of each Ca$^{2+}$ clasp becomes disordered (*Figure 3B*). 3D classification of the Ca$^{2+}$-free BEST1$_{345}$ dataset yielded only closed conformations of the neck but did indicate a degree of flexibility in the channel between the transmembrane and cytosolic regions that was manifested as a ~ 5° rotation along the symmetry axis (*Figure 3C*). This conformational flexibility was not observed in the Ca$^{2+}$-bound datasets, which suggests that Ca$^{2+}$ binding rigidifies the channel. We hypothesize that this Ca$^{2+}$-dependent rigidification may be necessary for opening of the neck.

## Coupling between the Ca$^{2+}$ clasp and the gate

When considering the amino acid side chain movements that are involved in the transition between the open and closed conformations of the neck and how Ca$^{2+}$ binding to the Ca$^{2+}$ clasp might bias these, our attention was drawn to the S4a helix, which is located behind the neck helix, and to W287 in particular. W287 is perfectly conserved among metazoan bestrophin channels and even conserved in the prokaryotic KpBEST channel (*Yang et al., 2014*) (*Figure 4A*); this sequence conservation would be consistent with an important role in channel gating. W287 is also positioned approximately at the midpoint between the Ca$^{2+}$ clasp and the neck (*Figure 2A,B*). W287 adopts one side chain rotamer to pack with F80 and F84 in the open conformation and a different side chain rotamer in the closed conformation that buttresses the space between adjacent neck helices (*Figure 2A,B*, *Figure 4E,F*). Because of its location and the conformational changes it undergoes, we hypothesized that W287 might propagate conformational changes in the Ca$^{2+}$ clasp to the neck and thereby govern the conformation of the neck. We find that the W287F mutation (BEST1$_{345}$ W287F) produces channels with dramatically altered gating. Whilst BEST1$_{345}$ W287F retained normal Cl$^-$ versus K$^+$ selectivity (*Figure 4B*), the W287F mutation constitutively activates the channel, making it nearly insensitive to the presence or absence of Ca$^{2+}$. Approximately, 80% of the Cl$^-$ current level through BEST1$_{345}$ W287F is maintained when Ca$^{2+}$ was chelated with EGTA, whereas currents through wild type BEST1 drop to zero (*Figure 4B,C*). To understand the molecular basis of this behavior, we determined cryo-EM structures of BEST1$_{345}$ W287F in the presence and absence of Ca$^{2+}$ at 2.7 Å

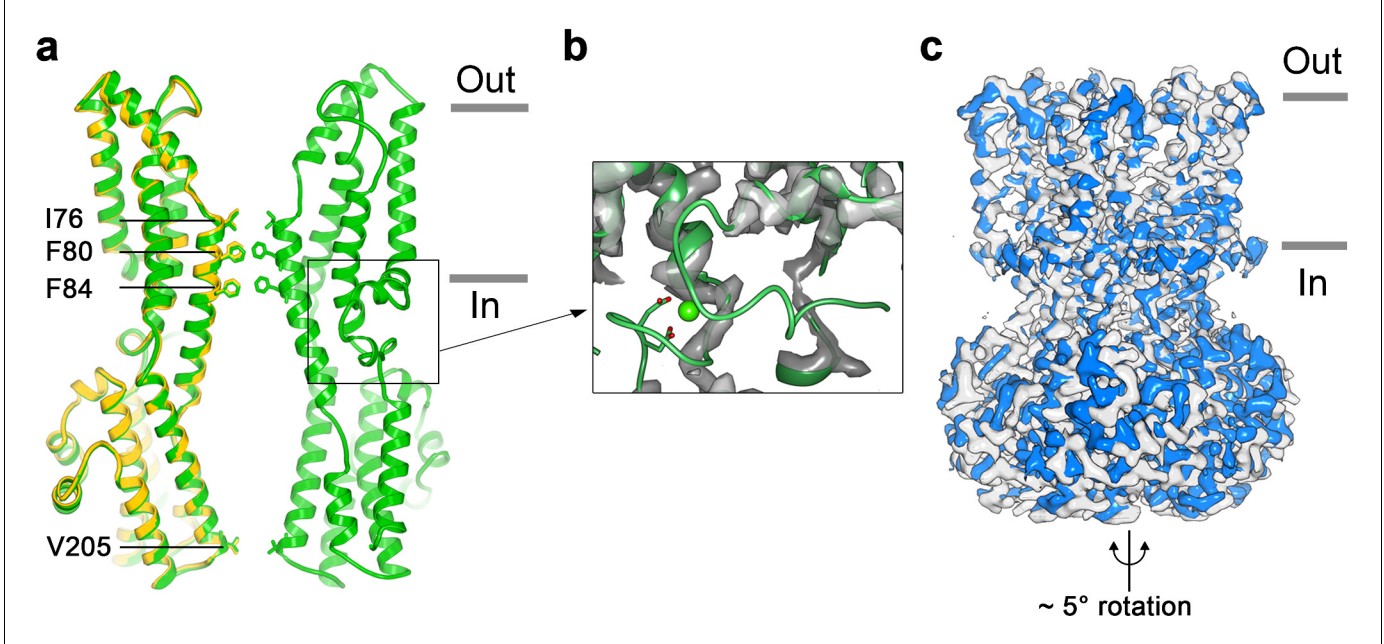

**Figure 3.** Structure of $Ca^{2+}$-free BEST1$_{345}$. (a) Overlay comparison of the $Ca^{2+}$-free conformation of BEST1$_{345}$ (yellow) with the $Ca^{2+}$-bound closed conformation of BEST1$_{345}$ (green). One ($Ca^{2+}$-free) or two ($Ca^{2+}$-bound) channel subunits in ribbon are shown from the side with the approximate boundaries of the bilayer indicated. The side chains of labeled residues are shown. The boxed area highlights the location of the $Ca^{2+}$-clasp. (b) Density for the $Ca^{2+}$-clasp is missing in the absence of $Ca^{2+}$. The structure of the $Ca^{2+}$-clasp region from the $Ca^{2+}$-bound closed conformation (green) is shown in comparison with the cryo-EM density in this region in the $Ca^{2+}$-free map, showing that the density for the $Ca^{2+}$ ion and surrounding protein residues are missing in the absence of $Ca^{2+}$. $Ca^{2+}$ is depicted as a green sphere and two aspartate residues that coordinate $Ca^{2+}$ as part of the $Ca^{2+}$ clasp are shown as sticks. (c) Refined cryo-EM maps of two conformations (blue, gray) of $Ca^{2+}$-free BEST1$_{345}$ that were identified using 3D classification. The maps are aligned according to their membrane-spanning regions, with the relative rotation between the cytosolic regions indicated.

DOI: https://doi.org/10.7554/eLife.43231.011

The following figure supplements are available for figure 3:

**Figure supplement 1.** Cryo-EM workflow for the BEST1$_{345}$ $Ca^{2+}$-free dataset.

DOI: https://doi.org/10.7554/eLife.43231.012

**Figure supplement 2.** Cryo-EM structure determination of: $Ca^{2+}$-free BEST1$_{345}$.

DOI: https://doi.org/10.7554/eLife.43231.013

**Figure supplement 3.** Map of $Ca^{2+}$-free BEST1$_{405}$.

DOI: https://doi.org/10.7554/eLife.43231.014

and 3.0 Å resolutions, respectively (*Figure 4—figure supplements 1* and *2*). In the presence of $Ca^{2+}$, BEST1$_{345}$ W287F adopts an open conformation that is essentially indistinguishable from the open conformation observed for BEST1$_{345}$ (*Figure 4—figure supplement 3, C*α RMSD = 0.2 Å). Unlike BEST1$_{345}$, 3D classification of the BEST1$_{345}$ W287F dataset did not reveal a closed conformation, which indicates that essentially all of the particles are in an open conformation and that the open state is preferential for this mutant. Remarkably, in the absence of $Ca^{2+}$, and in spite of missing density for $Ca^{2+}$ and a disordered $Ca^{2+}$-clasp region, the neck also adopts an open conformation (*Figure 4D*, *Figure 4—figure supplement 3*). Thus, in accord with the electrophysiological recordings, the W287F mutation decouples $Ca^{2+}$ binding from the conformational changes in the neck that are observed for the wild type channel. Modeling of the W287F mutation on a closed conformation of the channel introduces a void behind the neck (*Figure 4—figure supplement 3E*), which we hypothesize energetically disfavors the closed conformation. The effects of the relatively conservative mutation of this tryptophan to a phenylalanine give context to a myriad of disease-causing missense mutations in and around the neck (*Figure 2F*).

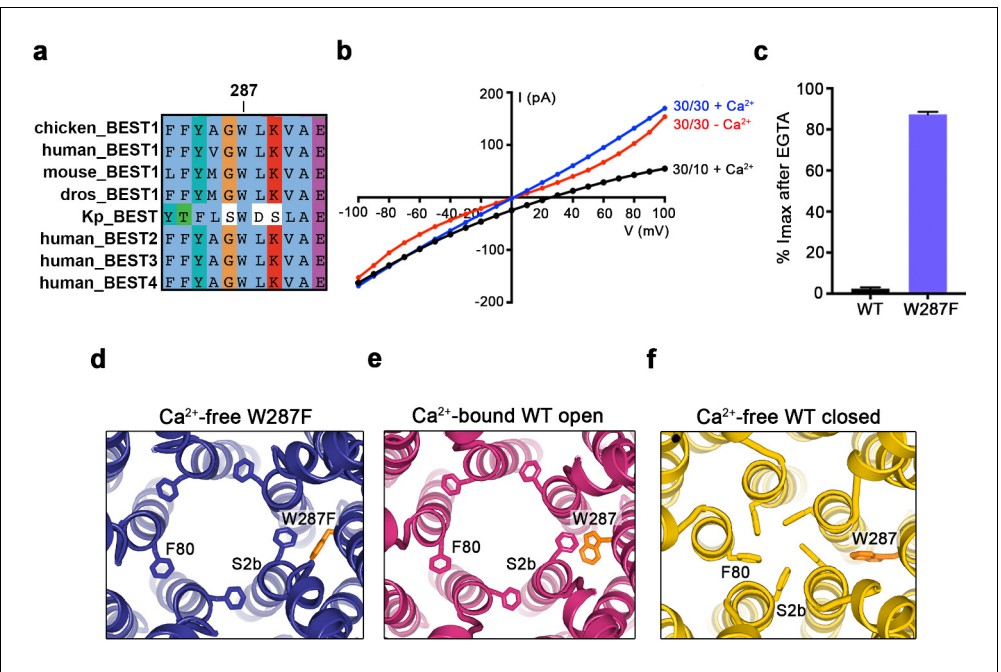

**Figure 4.** The W287F mutant decouples the $Ca^{2+}$ ligand from the activation gate. (**a**) Sequence alignment at and around residue 287 for chicken, human, mouse, *Drosophila melanogaster* (dros) and *Klebsiella pneumoniae* (Kp) bestrophin homologs and paralogs. (**b–c**) Dramatically reduced $Ca^{2+}$-dependence but normal $Cl^-$ versus $K^+$ selectivity of the W287F mutant. *I-V* relationships (**b**) are shown for voltages stepped from −100 to +100 mV for the indicated conditions [cis/trans KCl concentration in mM, and ~300 nM $[Ca^{2+}]_{free}$ (+$Ca^{2+}$) or 10 mM EGTA (-$Ca^{2+}$)]. The reversal potential ($E_{rev}$) measured using asymmetric KCl (30/10 mM) indicates normal $Cl^-$ versus $K^+$ selectivity: $E_{rev}$ = 24.8 ± 0.8 mV for BEST1$_{345}$ W287F in comparison to 23.4 ± 0.3 mV for wild type BEST1$_{345}$ (*Vaisey and Long, 2018*). (**c**) Bar graph showing the percentage of current remaining after addition of 10 mM EGTA for BEST1$_{345}$ (WT) and the W287F mutant. $I_{max}$ indicates the current measured at +100 mV in the presence of 300 nM $[Ca^{2+}]_{free}$. Error bars denote the SEM calculated from four (WT) or six (W287F) separate experiments. (**d– f**) The W287F mutant locks the neck open, even in the absence of $Ca^{2+}$. (**d**) Structure of the W287F mutant in the absence of $Ca^{2+}$, showing the open conformation of the neck region (ribbons; cutaway view from an extracellular orientation). The W287F mutation (orange sticks) is shown for one subunit. F80 residues are drawn as sticks. $Ca^{2+}$- bound open (**e**) and $Ca^{2+}$-free closed (**f**) conformations of BEST1$_{345}$ are depicted in the same manner.
DOI: https://doi.org/10.7554/eLife.43231.015

The following figure supplements are available for figure 4:

**Figure supplement 1.** Cryo-EM workflow for $Ca^{2+}$-bound and $Ca^{2+}$-free BEST1$_{345}$ W287F datasets.
DOI: https://doi.org/10.7554/eLife.43231.016

**Figure supplement 2.** Cryo-EM structure determination of $Ca^{2+}$-bound BEST1$_{345}$ W287F and $Ca^{2+}$-free BEST1$_{345}$ W287F.
DOI: https://doi.org/10.7554/eLife.43231.017

**Figure supplement 3.** Structure of $Ca^{2+}$-free BEST1$_{345}$ W287F.
DOI: https://doi.org/10.7554/eLife.43231.018

## The aperture

The structures reveal that the open pore of BEST1 comprises a 90 Å-long water-filled vestibule and a short constriction at the cytosolic aperture (*Figure 1D*). The aperture, which is located approximately 55 Å from the neck, is only ~3 Å long (measured where the pore diameter is <4 Å); its walls are formed solely by the V205 side chains of the five subunits (*Figure 5A*). Retinitis pigmentosa can be caused by mutation of the corresponding I205 residue of human BEST1 to threonine (*Davidson et al., 2009*). The aperture adopts the same conformation in all of the structures of BEST1 determined to date, regardless of whether the neck is open or closed or whether or not $Ca^{2+}$ is bound (*Figure 1C,D*, *Figure 3*, *Figure 4—figure supplement 3*). We have concluded previously from electrophysiological studies that the aperture does not function as either the activation or the

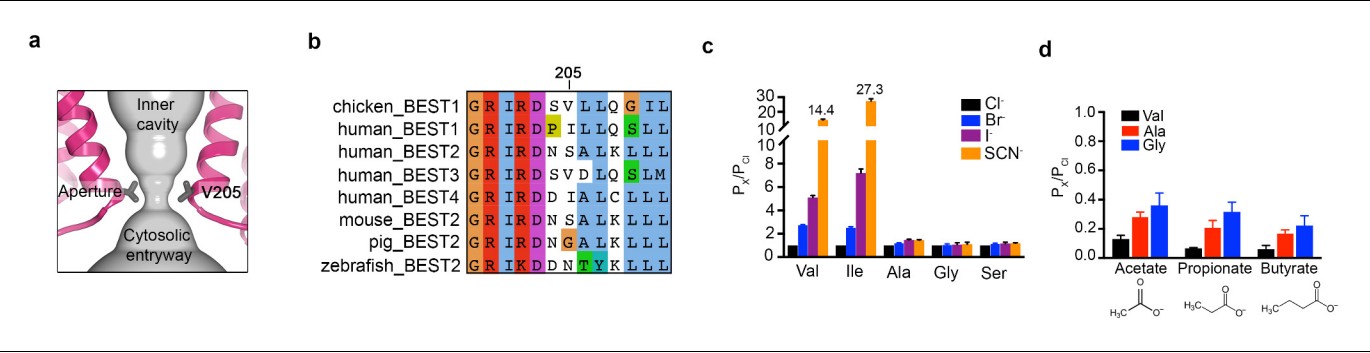

**Figure 5.** The aperture. (a) Close up of the aperture. (b) Sequence alignment around the aperture. (c–d) Mutation of V205 affects ion permeability. (c) Comparison of the permeabilities of Br⁻, I⁻, and SCN⁻ relative to Cl⁻ ($P_X/P_{Cl}$) for wild type (Val) and the indicated amino acid substitutions of V205. $P_X/P_{Cl}$ values were calculated from reversal potentials recorded in 30 mM KCl (cis) and 30 mM KX (trans) where X is Br, I, or SCN. The y-axis scale is discontinuous, as indicated, and the mean value of $P_{SCN}/P_{Cl}$ is indicated for the wild-type and the V205I mutant channels. IV traces are shown in *Figure 5—figure supplement 1*. (d) Permeabilities of larger anions. Comparison of the permeabilities of acetate, propionate and butyrate relative to Cl⁻ for wild type (Val) and the indicated mutants of V205 (calculated as in c). For (c–d), error bars denote the SEM from three experiments.

DOI: https://doi.org/10.7554/eLife.43231.019

The following figure supplement is available for figure 5:

**Figure supplement 1.** Current-voltage relationships of aperture mutants.
DOI: https://doi.org/10.7554/eLife.43231.020

inactivation gate (*Vaisey and Long, 2018*; *Vaisey et al., 2016*), and the structural data support this conclusion. Accordingly, the V205A mutation of chicken BEST1, which would be expected to widen the aperture markedly, has no effect on Ca²⁺-dependent activation or inactivation (*Vaisey and Long, 2018*; *Vaisey et al., 2016*). Mutations of V205 do, however, have dramatic effects on ion permeability (*Vaisey et al., 2016*) (*Figure 5C,D*, *Figure 5—figure supplement 1*). Both human and chicken BEST1 exhibit a lyotropic relative permeability sequence in which permeable anions that are more easily dehydrated than Cl⁻, such as Br⁻, I⁻ and SCN⁻, are more permeable than Cl⁻ (*Vaisey et al., 2016*; *Qu and Hartzell, 2000*; *Hartzell et al., 2005*; *Hille, 2001*) (*Figure 5C*, $P_X/P_{Cl}$ ~ 2.7 when X = Br, ~5.1 when X = I and ~14.4 when X = SCN). The lyotropic permeability sequence is an indication that at least partial dehydration occurs for ions as they permeate through the channel. Given the structural context of the open channel, the aperture is the only constriction where dehydration would occur (*Figure 1D,E*); the dimensions of the aperture are too narrow for a hydrated ion to pass and therefore the aperture would necessitate that anions shed at least some water molecules as they pass through it. The structures support the conclusion that the aperture is responsible for the lyotropic permeability sequence (*Vaisey et al., 2016*). Consistently, we find that mutation of V205 to a smaller residue (e.g. glycine, alanine, or serine) abolishes the lyotropic sequence among Cl⁻, Br⁻, I⁻ and SCN⁻ (*Figure 5C*), which suggests that the aperture is made sufficiently wide by these mutations to allow these ions to pass without dehydration. (It is worthwhile to note that our data are moot on the relative conductances among wild type and mutant channels but we hypothesize that wider apertures would yield larger conductances.) In contrast to the effects of small amino acids, mutation of V205 to isoleucine, a bulkier hydrophobic amino acid that would presumably reduce the diameter of the aperture, makes the lyotropic permeability differences between Cl⁻, Br⁻, I⁻ and SCN⁻ more dramatic (*Figure 5C*). These effects suggest that ions would shed more water molecules when passing through the V205I aperture than through the wild type one. Thus, the aperture controls permeability among halide anions by causing ion dehydration, and it thereby engenders the characteristic lyotropic permeability sequence of BEST1.

We also investigated the permeability of the larger organic anions acetate, propionate and butyrate relative to Cl⁻ for wild-type and aperture mutant channels (*Figure 5D*). These organic anions were less permeable than Cl⁻ (and less permeable than Br⁻, I⁻ and SCN⁻), which was expected due to their significantly larger size and the narrow dimensions of the aperture. We found that the relative permeability of the channel to acetate, propionate and butyrate increases when V205 is substituted

with the smaller alanine or glycine residues (*Figure 5D*). Thus, when the aperture is made wider by mutation, larger organic anions are more permeable. These data support the previous inference (*Vaisey et al., 2016*) that the aperture functions in the wild type channel as a size-selective filter. The aperture would tend to prevent the permeation of large cellular constituents such as proteins or nucleic acids but would allow smaller solutes such as $Cl^-$ ions to pass. However, the data indicate that the aperture is not perfectly discriminatory for the smallest solutes, as indicated by the measureable permeability of the wild type channel to acetate (*Figure 5D*). Notably, the amino acid sequence at and around amino acid 205 varies among human BEST1-4 channels and among bestrophin channels in general (*Figure 5B*). This sequence variability at the aperture may endow BEST1-4 channels with distinct permeabilities related to their specific physiological functions and their particular biological locations.

## Discussion

The structures presented herein represent the major gating transitions in the metazoan bestrophin channel. The work provides structural context to previous conclusions drawn from electrophysiological studies that the neck functions as both the activation and the inactivation gate (*Vaisey et al., 2016*; *Vaisey and Long, 2018*). *Figure 6* outlines our current understanding of the gating cycle. When unoccupied by $Ca^{2+}$, the $Ca^{2+}$ clasps are disordered, the neck is closed, and ions are prevented from permeating through the channel by the neck. When $Ca^{2+}$ ions bind, the $Ca^{2+}$ clasps become ordered and the open and closed conformations of the channel have free energies that are similar enough that the neck can sample both conformations. The open conformation of the neck is dramatically wider than the closed conformation and opening (activation) exposes small hydrophilic amino acids to the pore where large hydrophobic ones had been. The wide neck explains why the neck has essentially no influence on ion selectivity (*Vaisey et al., 2016*), in spite of speculation that it

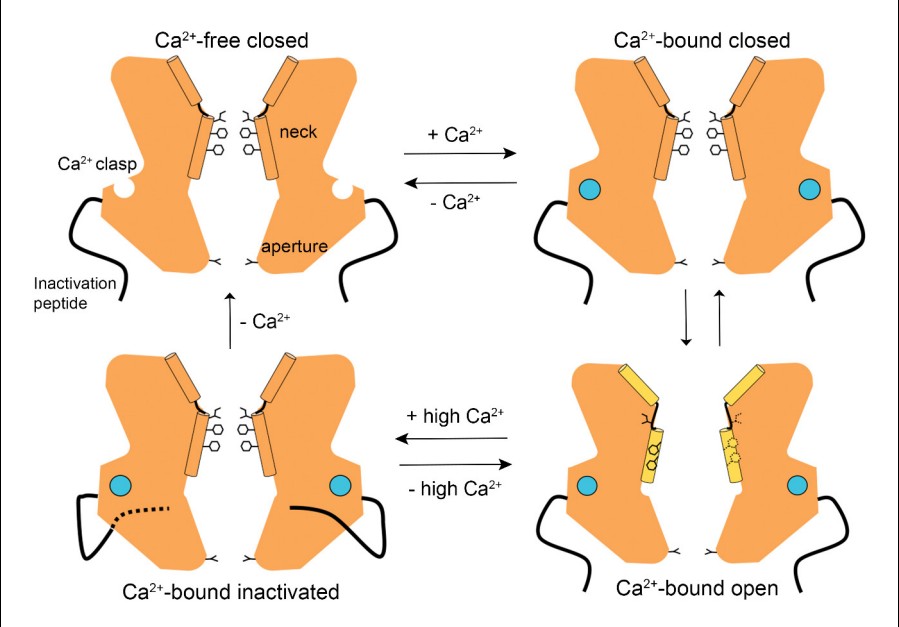

**Figure 6.** A Gating model. In the absence of $Ca^{2+}$, hydrophobic block at the neck prevents ion flow ($Ca^{2+}$-free closed). When the $Ca^{2+}$ clasps are occupied by $Ca^{2+}$, the channel is in equilibrium between $Ca^{2+}$-bound closed and $Ca^{2+}$-bound open conformations. The dramatic widening of the opened neck enables hydrated ions to flow through it. Binding of the inactivation peptide to its cytosolic receptor, which is stimulated by higher concentrations (>500 nM) of $Ca^{2+}$, induces the $Ca^{2+}$-bound inactivated conformation in which the neck is closed. The aperture, which remains fixed throughout the gating cycle, acts as a size-selective filter that requires permeating ions to become at least partially dehydrated as they pass though it, and this engenders the channel's lyotropic permeability sequence.
DOI: https://doi.org/10.7554/eLife.43231.021

might be involved in ion selectivity from the initial X-ray structure of BEST1 (*Kane Dickson et al., 2014*) and the structure of KpBest (*Yang et al., 2014*). The channel inactivates at higher concentrations of $Ca^{2+}$ through a mechanism that involves the binding of the C-terminal inactivation peptide to its receptor. The receptor is on the cytosolic surface of the channel and the binding of the inactivation peptide to it allosterically controls the conformation of the neck from a distance. The inactivated conformation of the neck is indistinguishable from the deactivated ($Ca^{2+}$-free) one.

Unlike voltage-dependent $K^+$ and $Na^+$ channels, in which ions are prevented from flowing by different mechanisms in the inactivated and deactivated states (*Hille, 2001*; *Armstrong and Bezanilla, 1977*), the same closed conformation of the neck is responsible for the deactivated and inactivated states of BEST1. Although this property in BEST1 is unusual among ion channels from a functional perspective, it seems less unusual from a structural perspective. Unlike many ion channels that have separate self-contained domains that are responsible for controlling channel gating, such as extracellular or intracellular ligand-binding domains, or voltage sensor domains within the membrane, which can often be studied in isolation using structural or functional approaches, BEST1 does not contain separable domains. The BEST1 channel comprises a single structural unit that resembles a barrel, with a portion of the barrel embedded in the membrane and a portion exposed to the cytosol. Additionally, the primary sequence of BEST1 does not have separate domains; rather the polypeptide portions that form the transmembrane and cytosolic regions of the channel are dispersed throughout the primary sequence (*Kane Dickson et al., 2014*). The midsection of the barrel contains the region that functions as a gate (the neck) and the cytosolic surface of the barrel is decorated with sensors that influence the gate – namely the $Ca^{2+}$ clasps and the inactivation peptide receptors. These sensors, which function as integral parts of the channel as a whole and could not be studied in isolation, modulate the energetics of opening and closing of the neck through allosteric means. The sensors do so by relaying information received at the surface of the channel, such as the binding of $Ca^{2+}$ or the inactivation peptide, through the core of the protein to the neck. The design of the neck itself appears well suited to receive this subtle energetic information in that the neck comprises five helices (the S2b neck helices) that are tethered on each end by linker regions that allow the helices to toggle between two preferred orientations that form the closed and open conformations of the neck. Thus, while localized twisting or domain motions that are instigated by separate sensor domains often constitute the activation mechanism of ion channels, in bestrophin cytosolic sensors on the surface of the channel control a dramatic molecular choreography within the core of the protein that underlies gating and reveals a fascinating new gating mechanism among ion channels.

Our electrophysiological and structural data suggest that the aperture remains essentially fixed throughout the $Ca^{2+}$-dependent activation and inactivation processes. Rather than functioning as a gate, as has been proposed (*Yang et al., 2014*), our data indicate that the aperture functions as a size-selective filter that discriminates permeant species on the basis of size and dehydration energy. Its dimensions and its hydrophobicity are such that any ion or hydrophilic entity passing through it must be at least partially dehydrated to fit. The aperture is thereby responsible for the lyotropic permeability sequence among small anions (e.g. $Cl^-$, $Br^-$, $I^-$, $SCN^-$) in which ions that are more easily dehydrated are more permeable through the channel. The aperture also limits the flow of larger anions such as acetate and mutations that increase the width of the aperture allow such ions to permeate more readily. Our data does not exclude the possibility that the aperture does change in dimensions somewhat due to thermal motion or due to some other factor. It has been observed, for example, that single channel currents obtained from purified KpBest are stimulated by ATP (*Zhang et al., 2018*). The mechanism of this regulation, and if it occurs in metazoan bestrophin channels, is not yet clear, but activation of bestrophins by ATP, which has been suggested to be mediated through changes at the aperture (*Zhang et al., 2018*), remains an intriguing possibility.

It has been proposed that the inhibitory neurotransmitter GABA permeates through BEST1 and underlies a tonic form of synaptic inhibition in glia (*Lee et al., 2010*). Previously, this possibility seemed incongruous with the narrowness of the neck observed in the X-ray structure (*Kane Dickson et al., 2014*) but the widened neck of the open channel conformation makes the possibility of slow conductance of GABA and/or other larger solutes more plausible from a structural perspective. While the dimensions of the aperture in the atomic models of BEST1, based on the cryo-EM and X-ray studies, would appear to preclude permeation of molecules the size of GABA, thermal motions that transiently widen the aperture may allow slow permeation of such solutes at physiologically relevant rates. Further, it is conceivable that regulatory mechanisms that influence the aperture exist and

would modulate such permeation. Given the importance of the aperture in regulating permeation, the sequence diversity in the aperture region is particularly intriguing. This diversity may give rise to distinct permeation properties among bestrophins and hints that the physiological functions of the channels are broader and more nuanced than currently appreciated.

## Materials and methods

### Cloning, expression and purification of BEST1

Chicken BEST1 (UniProt E1C3A0) constructs (amino acids 1–405 or 1–345 followed by a Glu-Gly-Glu-Glu-Phe tag) were expressed in *Pichia Pastoris* as described previously (*Kane Dickson et al., 2014*). Mutations were made using standard molecular biology techniques.

In preparation for cryo-EM analysis, purification of BEST1 proteins were performed as described previously with modification (*Kane Dickson et al., 2014*). BEST1 protein was purified by size-exclusion chromatography (SEC; Superose 6 increase 10/300 GL; GE Healthcare) in buffer containing 20 mM Tris, pH 7.5, 50 mM NaCl, 1 mM *n*-dodecyl-β-D-maltopyranoside (DDM; Anatrace) and 0.1 mM cholesteryl hemisuccinate (CHS; Anatrace). Purified BEST1 was concentrated to 5 mg ml$^{-1}$ using a 100 kDa concentrator (Amicon Ultra-4, Millipore) and divided into aliquots. For structures with $Ca^{2+}$, 1 μM $CaCl_2$ was added to the freshly purified protein. For $Ca^{2+}$-free structures, and 5 mM EGTA, pH 7.5 was added to the freshly purified protein. These samples were immediately used for cryo-EM grid preparation.

### EM sample preparation and data acquisition

5 μl of sample was pipetted onto Quantifoil R1.2/R1.3 holy carbon grids (Au 400, Electron Microscopy Sciences), which had been glow discharged for 10 s using a PELCO easiGlow glow discharge cleaning system (Ted Pella). A vitrobot Mark IV cryo-EM sample plunger (FEI) (operated at room temperature with a 1–2 s blotting time under a blot force of 0% and 100% humidity) was used to plunge-freeze the sample into liquid nitrogen-cooled liquid ethane. For $Ca^{2+}$-free conditions, the blotting paper used for grid freezing was pre-treated with 2 mM EGTA solution (4x), rinsed with $ddH_2O$ (4x) and dried under vacuum. Grids were clipped and loaded into a 300 keV Titan Krios microscope (FEI) equipped with a K2 Summit direct electron detector (Gatan). Images were recorded with SerialEM (*Mastronarde, 2005*) in super-resolution mode at a magnification of 22,500x, which corresponds to a super-resolution pixel size of 0.544 Å, and a defocus range of −0.7 to −2.15 μm. The dose rate was nine electrons per physical pixel per second, and images were recorded for 10 s with 0.25 s subframes (40 total frames), corresponding to a total dose of 76 electrons per Å$^2$.

### Image processing

*Figure 1—figure supplement 1*, *Figure 1—figure supplement 3*, *Figure 3—figure supplement 1* and *Figure 4—figure supplement 1* show the cryo-EM workflow for $Ca^{2+}$-bound BEST1$_{405}$, $Ca^{2+}$-bound BEST1$_{345}$, $Ca^{2+}$-free BEST1$_{345}$ and BEST1$_{345}$ W287F with and without $Ca^{2+}$, respectively. Movie stacks were gain-corrected, two-fold Fourier cropped to a calibrated pixel size of 1.088 Å, motion corrected and dose weighted using MotionCor2 (*Zheng et al., 2017*). Contrast Transfer Function (CTF) estimates for motion-corrected micrographs were performed in CTFFIND4 using all frames (*Rohou and Grigorieff, 2015*).

### $Ca^{2+}$-bound BEST1$_{405}$, BEST1$_{345}$ and BEST1$_{345}$ W287F datasets

All subsequent image processing was carried out with RELION2.1 (*Fernandez-Leiro and Scheres, 2017*), using a particle box size of 384 pixels and a spherical mask with a diameter of 140–160 Å. A total of 1740, 1644 and 1597 micrographs were collected for $Ca^{2+}$-bound BEST1$_{405}$, BEST1$_{345}$ and BEST1$_{345}$ W287F, respectively, and all micrographs were inspected manually; poor quality micrographs and those having CTF estimation fits lower than 5 Å were discarded. Approximately, 1000 particles were selected manually for reference-free 2D classification to generate templates that were then used for automatic particle picking. Auto-picking yielded ~312,000, ~309,000 and ~308,000 particles for BEST1$_{405}$, BEST1$_{345}$ and BEST1$_{345}$ W287F, respectively. One round of 2D classification, using 100 classes, was used to remove outlier particles (e.g. ice contaminants), and this

yielded ~290,000 particles for BEST1$_{405}$ and BEST1$_{345}$ datasets and ~265,000 particles for BEST1$_{345}$ W287F. 3D refinement, using C5 symmetry, was performed for each dataset using an initial model (generated from a previously collected, lower resolution cryo-EM dataset of Ca$^{2+}$-free BEST1 using EMAN2 [*Tang et al., 2007*]) that was low-pass filtered at 60 Å resolution. This yielded consensus reconstructions at 3.1 Å (BEST1$_{405}$)and 2.9 Å (BEST1$_{345}$) overall resolutions that have the closed conformation of the neck and 2.8 Å for BEST1$_{345}$ W287F that an open conformation at the neck. (Refinement using C1 symmetry also yielded reconstructions with 5-fold symmetry.) All overall resolution estimates are based on gold-standard Fourier shell correlations (FSC).

To identify the distinct conformational states within the Ca$^{2+}$-bound BEST1$_{345}$ dataset, we performed 3D classification using the consensus reconstruction as an initial model (low-pass filtered at 5 Å resolution) and sorting the particles into nine classes. One class with a widened neck (BEST1$_{345}$ open) was isolated, containing ~30,000 particles. To identify additional open particles from the dataset, this reconstruction was low-pass filtered at 5 Å resolution and used as an initial model for 3D classification (with four classes) on the entire dataset. This procedure yielded one class in the open conformation (containing approximately 40,000 particles) and three classes in the Ca$^{2+}$-bound closed conformation (containing the remainder of the particles). One class for the closed conformation was chosen (containing approximately 44,000 particles) because it contained better-resolved density for the residues lining the neck (I76, F80, F84). 3D Refinement of these two classes yielded reconstructions at 3.0 Å overall resolution. Particles from these two classes were 'polished' using aligned movie frames generated from MotionCor2 (*Zheng et al., 2017*). 3D refinement using the polished particles and a global angular sampling threshold of 1.75° yielded final reconstructions at 3.0 Å and 2.9 Å overall resolutions for the Ca$^{2+}$-bound closed and open reconstructions of BEST1$_{345}$, respectively. The same polishing strategy for the BEST1$_{345}$ W287F dataset yielded a final reconstruction of 2.7 Å. Several analogous 3D classification procedures were performed to try to identify an open conformation in the Ca$^{2+}$-bound BEST1$_{405}$ dataset but none were found. Conversely, 3D classification approaches with the Ca$^{2+}$-bound BEST1$_{W287F}$ dataset to identify multiple conformations yielded only reconstructions with an open neck.

## Ca$^{2+}$-free BEST1$_{345}$ and Ca$^{2+}$-free BEST1$_{345}$ W287F datasets

Initial image processing was carried out with RELION2.1 (*Fernandez-Leiro and Scheres, 2017*), using a particle box size of 384 pixels and a mask diameter of 140. A total of ~1000 and ~1600 micrographs were collected for Ca$^{2+}$-free BEST1$_{345}$ and Ca$^{2+}$-free BEST1$_{345}$ W287F, respectively, and manually pruned as described for the Ca$^{2+}$-bound dataset. Auto-picking templates were generated as described and the selected particles (~150,000 for the Ca$^{2+}$-free BEST1$_{345}$ and ~185,000 particles for the BEST1$_{345}$ W287F datasets) were subjected to one round of 2D classification with 100 classes. 3D refinement was performed using the selected particles from 2D classification (~130,000 for the Ca$^{2+}$-free BEST1$_{345}$ and ~150,000 particles for the Ca$^{2+}$-free BEST1$_{345}$ W287F datasets), C5 symmetry, and the EMAN2-generated initial model. This yielded reconstructions of 3.4 Å and 3.2 Å overall resolutions, respectively. Particle polishing was performed on each dataset and the polished particles were imported into the cisTEM cryo-EM software package for further refinement and classification (*Grant et al., 2018*). 3D refinement was performed in cisTEM using a mask to apply a 15 Å low-pass filter to the micelle region. This yielded final reconstructions to 3.0 Å overall resolution for both datasets. The ~5° relative rotation of the cytosolic region with respect to the transmembrane region that was observed under Ca$^{2+}$-free conditions was identified using 3D classification (using 6 and 8 classes for Ca$^{2+}$-free BEST1$_{345}$ and Ca$^{2+}$-free BEST1$_{345}$ W287F, respectively) using spatial frequencies up to 6 Å for refinement. Refinement of 3D classes with the most extreme rotation (e.g. approximately ±2.5 ° rotations relative to the consensus reconstruction) in cisTEM yielded overall resolutions of 3.6 Å (Ca$^{2+}$-free BEST1$_{345}$ conformation A, ~11,000 particles), 3.4 Å (Ca$^{2+}$-free BEST1$_{345}$ conformation B, ~21,000 particles), 3.4 Å (Ca$^{2+}$-free BEST1$_{345}$ W287F conformation A, ~21,000 particles) and 3.5 Å (Ca$^{2+}$-free BEST1$_{345}$ W287F conformation B, ~17,000 particles).

RELION2.1 (*Fernandez-Leiro and Scheres, 2017*) was used to estimate the local resolution of all the final maps. The maps shown in figures are combined maps, were sharpened (using a *B*-factor of −50–75 Å$^2$), and low-pass filtered at the final overall resolution of each map.

## Model building and refinement

The atomic models were manually built into one of the half-maps (which had been sharpened using a $B$-factor of $-50$–$75$ Å$^2$ and low-pass filtered at the final overall resolution) using the X-ray structure of BEST1 as a starting point (PDB ID: 4RDQ) and were refined in real space using the COOT software (*Emsley et al., 2010*). The atomic models were further refined in real space against the same half-map using PHENIX (*Adams et al., 2010*). The final models have good stereochemistry and good Fourier shell correlation with the other half-map as well as the combined map (*Figure 1—figure supplement 3*, *Figure 3—figure supplement 2, Figure 4—figure supplement 2*). Structural figures were prepared with Pymol (pymol.org), Chimera (*Pettersen et al., 2004*), and HOLE (*Smart et al., 1996*).

## Liposome reconstitution

SEC-purified protein [in SEC buffer: 150 mM NaCl, 20 mM Tris-HCl, pH7.5, 3 mM *n*-decyl-β-D-maltoside (DM; Anatrace)] was reconstituted into liposomes. A 3:1 (wt/wt) mixture of POPE (Avanti) and POPG (Avanti) lipids was prepared at 20 mg ml$^{-1}$ in reconstitution buffer (10 mM Hepes-NaOH, pH 7.6, 450 mM NaCl, 0.2 mM EGTA, 0.19 mM CaCl$_2$). 8% (wt/vol) *n*-octyl-β-D-maltopyranoside (Anatrace) was added to solubilize the lipids, and the mixture was incubated with rotation for 30 min at room temperature. Purified protein was mixed with an equal volume of the solubilized lipids to give a final protein concentration of 0.2–1 mg ml$^{-1}$ and a lipid concentration of 10 mg ml$^{-1}$. Proteoliposomes were formed by dialysis (using a 8000 Da molecular mass cutoff) for 1–2 days at 4°C against 2–4 L of reconstitution buffer and were flash frozen in liquid nitrogen and stored at $-80^{°C}$ until use.

## Electrophysiological recordings

Proteoliposomes were thawed and sonicated for approximately 10 s using an Ultrasonic Cleaner (Laboratory Supplies Company). All data are from recordings made using the Warner planar lipid bilayer workstation (Warner Instruments). Two aqueous chambers (4 mL) were filled with bath solutions. Chlorided silver (Ag/AgCl) wires were used as electrodes, submerged in 3 M KCl, and connected to the bath solutions via agar-KCl salt bridges [2% (wt/vol) agar, 3 M KCl]. The bath solutions were separated by a polystyrene partition with a ~ 200 µM hole across which a bilayer was painted using POPE:POPG in *n*-decane [3:1 (wt/wt) ratio at 20 mg ml$^{-1}$]. Proteoliposomes were applied to the bilayer with an osmotic gradient across the bilayer with solutions consisting of: 30 mM KCl or NaCl (*cis* side) and 10 mM KCl or NaCl (*trans* side), 20 mM Hepes-NaOH, pH 7.6 and 0.21 mM EGTA/0.19 mM CaCl$_2$ ([Ca$^{2+}$]$_{free}$ ~300 nM) or 1 µM CaCl$_2$. Proteoliposomes were added, 1 µL at a time, to the *cis* chamber to a preformed bilayer until ionic currents were observed. Solutions were stirred using a stir plate (Warner Instruments stir plate) to aid vesicle fusion. After fusion, the solutions were made symmetric by adding 3M KCl or 5M NaCl, depending on the starting solutions, to the *trans* side. Unless noted, all reagents were purchased from Sigma-Aldrich. All electrical recordings were taken at room temperature (22–24°C).

Measurements of relative permeabilities among anions were performed as described previously (*Vaisey et al., 2016*). Briefly, after establishing symmetric (30/30 mM KCl or NaCl) conditions, the bath solution in the *trans* chamber was replaced by perfusion with solutions in which KCl or NaCl was replaced by various potassium salts (Br, I, SCN, acetate, propionate) or sodium salts (butyrate).

Currents were recorded using the Clampex 10.4 program (Axon Instruments) with an Axopatch 200B amplifier (Axon Instruments) and were sampled at 200 µs and filtered at 1 kHz. Data were analyzed using Clampfit 10.4 (Axon Instruments). Graphical display and statistical analyses were carried out using GraphPad Prism 6.0 software. In all cases, currents from bilayers without channels were subtracted. Error bars represent the SEM of at least three separate experiments, each in a separate bilayer. We define the side to which the vesicles are added as the *cis* side and the opposite *trans* side as electrical ground, so that transmembrane voltage is reported as V$_{cis}$-V$_{trans}$. Ion channels are inserted in both orientations in the bilayer.

## Acknowledgements

We thank N Grigorieff, members of his laboratory, and the staff at the Howard Hughes Medical Institute Cryo-EM facility for training and initial advice on cryo-EM. We thank MJ de la Cruz of the

Memorial Sloan Kettering Cancer Center Cryo-EM facility, M Ebrahim, and the staff of the New York Structural Biology Center Simons Electron Microscopy Center for help with data collection. G.V received funding and mentorship from the Boehringer Ingelheim Fonds Predoctoral Fellowship Program. Funding: This work was supported, in part, by NIH Grant R01 GM110396 (to SBL) and a core facilities support grant to Memorial Sloan Kettering Cancer Center (P30 CA008748).

## Additional information

### Funding

| Funder | Grant reference number | Author |
|---|---|---|
| National Institutes of Health | R01-GM110396 | Stephen B Long |
| National Cancer Institute | P30 CA008748 | Stephen B Long |

The funders had no role in study design, data collection and interpretation, or the decision to submit the work for publication.

### Author contributions

Alexandria N Miller, Conceptualization, Data curation, Formal analysis, Validation, Visualization, Methodology, Writing—original draft, Writing—review and editing, Determined structures of BEST1405, the W287F mutant, and structures in the absence of Ca2+, Performed electrophysiology experiments; George Vaisey, Conceptualization, Data curation, Formal analysis, Validation, Visualization, Methodology, Writing—original draft, Writing—review and editing, Determined structures of the Ca2+-bound closed state and the Ca2+-bound open state, Performed electrophysiology experiments; Stephen B Long, Conceptualization, Supervision, Funding acquisition, Validation, Visualization, Methodology, Project administration, Writing—review and editing

### Author ORCIDs

George Vaisey (iD) http://orcid.org/0000-0002-8359-1314
Stephen B Long (iD) http://orcid.org/0000-0002-8144-1398

### Decision letter and Author response

Decision letter https://doi.org/10.7554/eLife.43231.048
Author response https://doi.org/10.7554/eLife.43231.049

## Additional files

### Supplementary files

• Transparent reporting form
DOI: https://doi.org/10.7554/eLife.43231.022

### Data availability

Atomic coordinates and cryo-EM density maps of have been deposited with the PDB and Electron Microscopy Data Bank with the accession numbers: 6N23 (BEST1405, inactivated; EMD-9321), 6N24 (BEST1345 W287F mutant, Ca2+-free; EMD-9322), 6N25 (BEST1345 W287F mutant, Ca2+-bound; EMD-9323), 6N26 ( BEST1345 Ca2+-free closed state; EMD-9324), 6N27 (BEST1345 Ca2+-bound closed state; EMD-9325), and 6N28 ( BEST1345 Ca2+-bound open state; EMD-9326).

The following datasets were generated:

| Author(s) | Year | Dataset title | Dataset URL | Database and Identifier |
|---|---|---|---|---|
| Miller AN, Vaisey G, Long SB | 2018 | BEST1 in a calcium-bound inactivated state | https://www.rcsb.org/structure/6N23 | Protein Databank, 6N23 |
| Miller AN, Vaisey G, Long SB | 2018 | BEST1 open state W287F mutant, calcium-free | https://www.rcsb.org/structure/6N24 | Protein Databank, 6N24 |

| Miller AN, Vaisey G, Long SB | 2018 | BEST1 open state W287F mutant, calcium-bound | https://www.rcsb.org/structure/6N25 | Protein Databank, 6N25 |
|---|---|---|---|---|
| Miller AN, Vaisey G, Long SB | 2018 | BEST1 calcium-free closed state (deactivated) | https://www.rcsb.org/structure/6N26 | Protein Databank, 6N26 |
| Miller AN, Vaisey G, Long SB | 2018 | BEST1 calcium-bound closed state | https://www.rcsb.org/structure/6N27 | Protein Databank, 6N27 |
| Miller AN, Vaisey G, Long SB | 2018 | BEST1 calcium-bound open state | https://www.rcsb.org/structure/6N28 | Protein Databank, 6N28 |
| Miller AN, Vaisey G, Long SB | 2018 | BEST1 in a calcium-bound inactivated state | https://www.ebi.ac.uk/pdbe/entry/emdb/EMD-9321 | Electron Microscopy Data Bank, EMD-9321 |
| Miller AN, Vaisey G, Long SB | 2018 | BEST1 open state W287F mutant, calcium-free | https://www.ebi.ac.uk/pdbe/entry/emdb/EMD-9322 | Electron Microscopy Data Bank, EMD-9322 |
| Miller AN, Vaisey G, Long SB | 2018 | BEST1 open state W287F mutant, calcium-bound | https://www.ebi.ac.uk/pdbe/entry/emdb/EMD-9323 | Electron Microscopy Data Bank, EMD-9323 |
| Miller AN, Vaisey G, Long SB | 2018 | BEST1 calcium-free closed state (deactivated) | https://www.ebi.ac.uk/pdbe/entry/emdb/EMD-9324 | Electron Microscopy Data Bank, EMD-9324 |
| Miller AN, Vaisey G | 2018 | BEST1 calcium-bound closed state | https://www.ebi.ac.uk/pdbe/entry/emdb/EMD-9325 | Electron Microscopy Data Bank, EMD-9325 |
| Miller AN, Vaisey G, Long SB | 2018 | BEST1 calcium-bound open state | https://www.ebi.ac.uk/pdbe/entry/emdb/EMD-9326 | Electron Microscopy Data Bank, EMD-9326 |

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
