## [Decision Letter]

Thank you for submitting your article "Molecular mechanisms of gating in the calcium-activated chloride channel bestrophin" for consideration by *eLife*. Your article has been reviewed by three peer reviewers, and the evaluation has been overseen by Kenton Swartz as the Reviewing Editor and Richard Aldrich as the Senior Editor. The following individuals involved in review of your submission have agreed to reveal their identity: Joel Meyerson (Reviewer #1); Christopher Miller (Reviewer #2); H Criss Hartzell (Reviewer #3).

The reviewers have discussed the reviews with one another and the Reviewing Editor has drafted this decision to help you prepare a revised submission.

Summary:

This very nice manuscript from the Long laboratory presents new insights into the gating and permeation properties of the BEST1 Ca^2+^-activated Cl^-^ channel. This research is significant not only because mutations in BEST1 are linked to a spectrum of inherited retinal dystrophies, but also because the mechanisms of gating of this channel by Ca^2+^ have remained obscure despite extensive mutagenesis and structural studies. In this paper, the Long lab has solved the cryo-EM structures of chicken BEST1 in several conformations that reveal the mechanisms of Ca^2+^-dependent gating as well as some insights into the mechanisms of anion selectivity. The authors show that opening of the BEST1 channel involves a dilation of the neck region of the pore that occurs because hydrophobic residues I76, F80, and F84 move out of the pore in a concerted manner and repack with residues located in surrounding non-pore lining helices. The authors also show that the aperture of the pore is likely to form the anionic selectivity filter. We have no major concerns with what can only be described as a wonderful manuscript: it is clearly written, and the data are compelling. The following are several issues that we would like to have addressed.

1) Subsection “Opening transition of the neck”. While we think that we understand the basic mechanics of gating presented here, one reviewer had trouble visualizing the "impressive flexibility" and "tethering" that are provided by "disruptions" of the α-helix that allow the helix to "float". One might argue that the imagery of the wording clouds, rather than explains, what is happening. The more precise description that follows is much clearer and could stand alone. Also, you might consider a description that begins with the assumption that gating is initiated by Ca^2+^ stabilizing the "open" conformation of helix S4a and explaining how this conformational change may lead to the changes in the gating residues. For example, the sentence "In a domino effect", makes it sound like the movements of F282, F283, and W287 are a consequence of the movements of F80 and F84, which seems backwards. Also, in the closed conformation, Q280, E74, and Y72 are fairly close together and we wondered whether Ca^2+^ binding disrupts some polar interactions between S2b and S4a that stabilize the closed state. Also, in this regard, the description of the role of W287 could be clarified and refined.

2) Subsection “The aperture”. The finding that mutation of V205 to more hydrophilic residues abolishes the lyotropic sequence while replacing with a bulkier hydrophobic amino acid enhances the permeability differences deserves more attention. One might expect that replacing V205 with smaller more hydrophilic residues would increase the conductance of larger anions that are dehydrated more easily. But, the V205A/G/S mutations apparently reduce the conductance of all anions with larger anions being affected more than smaller anions. Do the authors have a model to explain the changes in permeability with these mutations or are these mutations significantly distorting the structure of the aperture? Could the authors also show bar graphs of G_x_/G_Cl_? One might also worry about the accuracy of permeability measurements when outward currents are so small. Could outward currents in Figure 5C, D be carried by K^+^?

3) Discussion, last paragraph. The dimensions of the aperture of cBEST1 would seem to preclude the permeation of large molecules. We presume the statement that there is "variability in the aperture" refers to primary sequence, not structure. While we like the suggestion that BESTs may have some unexpected functions, this paragraph should be more rigorously constructed or deleted.

4) Introduction, first paragraph. Should also cite Fischmeister and Hartzell (2005).

5) What is an "indistinguishable closed conformation"?

6) Subsection “The aperture”: *Drosophila* should be capitalized.

7) Discussion, third paragraph. What is the reasoning that led the authors to suggest that ATP may gate the channel via the aperture?

8) Results, first paragraph: The line reads "…resolution cryo-EM structure of BEST1…". Readability would be improved if here the text indicated that the structure is calcium bound.

9) "W287 is highly conserved among bestrophin channels." A sequence alignment to show the conservation would improve readability. This could simply be added as a supplemental panel.

10) "... cryo-EM structures of the W287F mutant…". Given the large number of structures in the paper, minor readability improvement could be gained by indicating this is the BEST1_345_ construct being discussed.

11) "…residues I76 (e), F80 (f) and F84 (h)" should be "…residues I76 (b), F80 (c) and F84 (d)"

12) Figure 1: Here the authors very nicely present the difference between the closed and open pore models, as a cutaway with two subunits. Given how important this classification result is to the manuscript, the presentation would benefit from showing a similar cutaway of the pore opening, but with the cryo-EM maps together with the models. This could be done in a small supplemental panel.

13) Minor typos: "In Ca^2+^..." should be "In the presence of Ca^2+^…"

14) "A total of ~1000 or ~1600 micrographs were collected for Ca^2+^-free BEST1_345_ and Ca^2+^-free BEST1_345_ W287F, respectively […]" should be "A total of ~1000 and ~1600 micrographs were collected for Ca^2+^-free BEST1_345_ and Ca^2+^-free BEST1_345_ W287F, respectively […]"

15) "(using 6 or 8 classes for Ca^2+^-free BEST1_345_ and Ca^2+^-free BEST1_345_ W287F, respectively)" should be "(using 6 and 8 classes for Ca^2+^-free BEST1_345_ and Ca^2+^-free BEST1_345_ W287F, respectively)"

16) Figure 4—figure supplement 3. "…Structure of Ca^2+^-free BEST1W287F should be "...Structure of Ca^2+^-free BEST1_345_ W287F "

17) A few lines where spaces are needed"

"…do so my relaying…" should be "…do so by relaying…"

"previously(Kane Dickson" should be "previously (Kane Dickson"

"SerialEM(Mastronarde" should be "SerialEM (Mastronarde"

"RELION2.1(Fernandez-Leiro" should be "RELION2.1 (Fernandez-Leiro"

"MotionCor2(Zheng" should be "MotionCor2 (Zheng"

"previously(Vaisey" should be "previously (Vaisey"

Figure 1: first panel reads "309, 189" rather than "309,189"

18) In one of the citations of the lyotropic permeability sequence in this channel, it would be useful to quote the actual permeability ratios to Cl^-^ for WT BEST. (Figure 5 is hard to read for WT – and oh, where is the always-outcast poor cousin F?).

19) Subsection “The aperture”: Remind readers how far the aperture is from the neck in Å.

20) Subsection “An open channel conformation”, last paragraph. We don't agree that Raoo et al. – a purely computational study – is evidence for 'hydrophobic block' – a mechanism that carries a strong connotation that a vacuum exists in that region (as originally defined by Sansom). Equally possible (or more plausible) is that water – invisible to you – occupies that space, which is simply a high kinetic barrier to mostly dehydrated Cl^-^. You might better refer to this MD study as being consistent with hydrophobic block, but phrased as now, it strikes us as too strong.

---

## [Author Response]

Minor points:1) Subsection “Opening transition of the neck”. While we think that we understand the basic mechanics of gating presented here, one reviewer had trouble visualizing the "impressive flexibility" and "tethering" that are provided by "disruptions" of the α-helix that allow the helix to "float". One might argue that the imagery of the wording clouds, rather than explains, what is happening. The more precise description that follows is much clearer and could stand alone. Also, you might consider a description that begins with the assumption that gating is initiated by Ca^2+^ stabilizing the "open" conformation of helix S4a and explaining how this conformational change may lead to the changes in the gating residues. For example, the sentence "In a domino effect", makes it sound like the movements of F282, F283, and W287 are a consequence of the movements of F80 and F84, which seems backwards. Also, in the closed conformation, Q280, E74, and Y72 are fairly close together and we wondered whether Ca^2+^ binding disrupts some polar interactions between S2b and S4a that stabilize the closed state. Also, in this regard, the description of the role of W287 could be clarified and refined.

Thank you for these comments. In the revised manuscript, we have attempted to clarify the description of the conformational changes. We have removed the language of “domino effect” and “float”. Thank you for mentioning the intriguing possibility that polar interactions between S2b and S4a contribute to the stability of the closed state. The gating transition undoubtedly involves multiple energetic factors but we prefer to refrain from speculating further in this regard. The existence of both closed and open conformations of the neck when Ca^2+^ is bound indicates that the gating transition is governed by relatively small energy differences once Ca^2+^ is bound.

2) Subsection “The aperture”. The finding that mutation of V205 to more hydrophilic residues abolishes the lyotropic sequence while replacing with a bulkier hydrophobic amino acid enhances the permeability differences deserves more attention. One might expect that replacing V205 with smaller more hydrophilic residues would increase the conductance of larger anions that are dehydrated more easily. But, the V205A/G/S mutations apparently reduce the conductance of all anions with larger anions being affected more than smaller anions. Do the authors have a model to explain the changes in permeability with these mutations or are these mutations significantly distorting the structure of the aperture? Could the authors also show bar graphs of G_x_/G_Cl_? One might also worry about the accuracy of permeability measurements when outward currents are so small. Could outward currents in Figure 5C, D be carried by K^+^?

Thank you for your comments. We have clarified possible mechanisms for the effects of mutations to V205 on the relative permeabilities. We do not expect mutation of V205 to A/S/G/I to cause major distortions to the structure of the protein.

Regarding relative conductances, we now include a bar chart showing the relative conductance of SCN^-^ to Cl^-^ for the wild type channel and the aperture mutants (Figure 5—figure supplement 1C). The trend is consistent with the changes in relative ion permeabilities, which we do not find surprising. Mutations to smaller amino acids (e.g. G/S/A) reduce the relative difference in conductance between larger ions and Cl^-^ that is observed for the wild-type channel, and this is similar to what we observe for the relative ion permeabilities. Because the electrophysiological studies are of macroscopic currents from an undetermined number of channels, we do not know if the mutations reduce or increase overall channel conductance (and/or change the open probability). Therefore, we submit that the reviewer is not correct to conclude that “the V205A/G/S mutations apparently reduce the conductance of all anions with larger anions being affected more than smaller anions.” Anecdotally, we find that smaller amino acid substitutions of the aperture residue (G/S/A) tend to yield larger macroscopic currents than the wild type channel for a given amount of proteoliposomes (this may indicate an increased overall channel conductance for these mutants); consequently, we typically used lower amounts of proteoliposomes for the planar lipid bilayer measurements of these mutants.

We think that it is unlikely that the outward currents observed in Figure 5—figure supplement 1D and E are carried by K^+^. Although the outward currents observed in Figure 5—figure supplement 1D and E are small, the measured reversal potentials were consistent for a given mutation and different mutants yielded different reversal potentials (as indicated in Figure 5D). These differences in reversal potentials are not due to differences in relative permeabilities of K^+^ (or Na^+^) to Cl^-^ among the mutants, which would have been apparent from the experiments shown in Figure 5—figure supplement 1 A.

3) Discussion, last paragraph. The dimensions of the aperture of cBEST1 would seem to preclude the permeation of large molecules. We presume the statement that there is "variability in the aperture" refers to primary sequence, not structure. While we like the suggestion that BESTs may have some unexpected functions, this paragraph should be more rigorously constructed or deleted.

Thank you for your comments. We have re-crafted this paragraph and clarified that our reference to variability was with regard to primary sequence.

4) Introduction, first paragraph. Should also cite Fischmeister and Hartzell (2005).

Thank you – we have included this citation.

5)What is an "indistinguishable closed conformation"?

Thank you. We agree that the previous phrasing was confusing and have clarified that the neck adopts the same conformation in the Ca^2+^-bound closed and inactivated structures.

6) Subsection “The aperture”: Drosophila should be capitalized.

Thank you. We have made this correction.

7) Discussion, third paragraph. What is the reasoning that led the authors to suggest that ATP may gate the channel via the aperture?

Thank you for your comment. We have clarified this hypothesis.

8) Results, first paragraph: The line reads "…resolution cryo-EM structure of BEST1…". Readability would be improved if here the text indicated that the structure is calcium bound.

Thank you for this suggestion. We have made this change.

9) "W287 is highly conserved among bestrophin channels." A sequence alignment to show the conservation would improve readability. This could simply be added as a supplemental panel.

Thank you for this suggestion. We have added a sequence alignment to Figure 4.

10) "… cryo-EM structures of the W287F mutant…". Given the large number of structures in the paper, minor readability improvement could be gained by indicating this is the BEST1_345_ construct being discussed.

Thank you for this suggestion. We have made this change.

11) "…residues I76 (e), F80 (f) and F84 (h)" should be "…residues I76 (b), F80 (c) and F84 (d)"

Thank you for this correction.

12) Figure 1: Here the authors very nicely present the difference between the closed and open pore models, as a cutaway with two subunits. Given how important this classification result is to the manuscript, the presentation would benefit from showing a similar cutaway of the pore opening, but with the cryo-EM maps together with the models. This could be done in a small supplemental panel.

Thank you for this suggestion. We have included additional views of the cryo-EM maps in Figure 1 and Figure 1—figure supplement 4.

13) Minor typos: "In Ca^2+^,…" should be "In the presence of Ca^2+^,…"14) "A total of ~1000 or ~1600 micrographs were collected for Ca^2+^-free BEST1_345_ and Ca^2+^-free BEST1_345_ W287F, respectively […]" should be "A total of ~1000 and ~1600 micrographs were collected for Ca^2+^-free BEST1_345_ and Ca^2+^-free BEST1_345_ W287F, respectively […]"15) "(using 6 or 8 classes for Ca^2+^-free BEST1_345_ and Ca^2+^-free BEST1_345_ W287F, respectively)" should be "(using 6 and 8 classes for Ca^2+^-free BEST1_345_ and Ca^2+^-free BEST1_345_ W287F, respectively)"16) Figure 4—figure supplement 3. "…Structure of Ca^2+^-free BEST1W287F " should be "...Structure of Ca^2+^-free BEST1_345_ W287F "17) A few lines where spaces are needed""…do so my relaying…" should be "…do so by relaying…""previously(Kane Dickson" should be "previously (Kane Dickson""SerialEM(Mastronarde" should be "SerialEM (Mastronarde""RELION2.1(Fernandez-Leiro" should be "RELION2.1 (Fernandez-Leiro""MotionCor2(Zheng" should be "MotionCor2 (Zheng""previously(Vaisey" should be "previously (Vaisey"Figure 1: first panel reads "309, 189" rather than "309,189"18) In one of the citations of the lyotropic permeability sequence in this channel, it would be useful to quote the actual permeability ratios to Cl^-^ for WT Best. (Figure 5 is hard to read for WT – and oh, where is the always-outcast poor cousin F?).19) Subsection “The aperture”: Remind readers how far the aperture is from the neck in Å.

Thank you. We have made these corrections.

20) Subsection “An open channel conformation”, last paragraph. We don't agree that Raoo et al. – a purely computational study – is evidence for 'hydrophobic block' – a mechanism that carries a strong connotation that a vacuum exists in that region (as originally defined by Sansom). Equally possible (or more plausible) is that water – invisible to you – occupies that space, which is simply a high kinetic barrier to mostly dehydrated Cl^-^. You might better refer to this MD study as being consistent with hydrophobic block, but phrased as now, it strikes us as too strong.

Thank you for these comments. We have altered the phrasing to refer to an “energetic barrier” that occurs at the neck.